# Pore-forming alpha-hemolysin efficiently improves the immunogenicity and protective efficacy of protein antigens

Jin-Tao Zou[1], Hai-Ming Jing[1], Yue Yuan[1], Lang-Huan Lei[1,2], Zhi-Fu Chen[1], Qiang Gou[1], Qing-Shan Xiong[1], Xiao-Li Zhang[3], Zhuo Zhao[1], Xiao-Kai Zhang[1], Hao Zeng[1], Quan-Ming Zou[1]*, Jin-Yong Zhang[1]*

1 National Engineering Research Center of Immunological Products & Department of Microbiology and Biochemical Pharmacy, College of Pharmacy, Third Military Medical University, Chongqing, PR China, 2 Department of Critical Care Medicine, Children's Hospital of Chongqing Medical University, Chongqing, PR China, 3 Department of Clinical Hematology, College of Pharmacy, Third Military Medical University, Chongqing, PR China

* qmzou2007@163.com (Q-MZ); zhangjy198217@126.com (J-YZ)

**Data Availability Statement:** Raw data files for RNA-seq have been deposited in the NCBI Gene Expression Omnibus under accession number GEO: GSE167077. All other relevant data are within

## Abstract

Highly immunogenic exotoxins are used as carrier proteins because they efficiently improve the immunogenicity of polysaccharides. However, their efficiency with protein antigens remains unclear. In the current study, the candidate antigen PA0833 from *Pseudomonas aeruginosa* was fused to the α-hemolysin mutant Hla$^{H35A}$ from *Staphylococcus aureus* to form a Hla$^{H35A}$-PA0833 fusion protein (HPF). Immunization with HPF resulted in increased PA0833-specific antibody titers, higher protective efficacy, and decreased bacterial burden and pro-inflammatory cytokine secretion compared with PA0833 immunization alone. Using fluorescently labeled antigens to track antigen uptake and delivery, we found that Hla$^{H35A}$ fusion significantly improved antigen uptake in injected muscles and antigen delivery to draining lymph nodes. Both *in vivo* and *in vitro* studies demonstrated that the increased antigen uptake after immunization with HPF was mainly due to monocyte- and macrophage-dependent macropinocytosis, which was probably the result of HPF binding to ADAM10, the Hla host receptor. Furthermore, a transcriptome analysis showed that several immune signaling pathways were activated by HPF, shedding light on the mechanism whereby Hla$^{H35A}$ fusion improves immunogenicity. Finally, the improvement in immunogenicity by Hla$^{H35A}$ fusion was also confirmed with two other antigens, GlnH from *Klebsiella pneumoniae* and the model antigen OVA, indicating that Hla$^{H35A}$ could serve as a universal carrier protein to improve the immunogenicity of protein antigens.

## Author summary

Pore-forming toxins, a kind of exotoxins utilized by many pathogens as immune escaping weapons that targeting the immune cells and disturbing the immune system, are conventionally deemed as perfect antigens for vaccine development against infectious diseases. In this study, we reported that fusion of Hla$^{H35A}$, a typical pore-forming toxin toxoid, to

the manuscript and its Supporting Information files.

**Funding:** This work was funded by the National Natural Science Foundation of China (grant number 31970138) to JYZ. The funders had no role in study design, data collection and analysis, decision to publish, or preparation of the manuscript.

**Competing interests:** The authors have declared that no competing interests exist.

candidate protein antigens from different species resulted in improved immunogenicity and protective efficacy. The improvement was mainly due to the increased antigen uptake and activating of immune-associated signaling pathways, probably by targeting ADAM10, the receptor of Hla on host immune cells. The importance of this work was to demonstrate the possible mechanisms of that pore-forming toxin function as atypical carrier protein to improve the immunogenicity of other proteins and confirm the potential of non-conservative but highly immunogenic exotoxins derived from hyper-virulent clinical strains for application in rational antigen design and vaccines development.

## Introduction

Antibiotic-resistant infections have become an urgent global threat to public health [1]. In 2017, the World Health Organization (WHO) released a list of drug-resistant bacteria, which consists of 12 'priority pathogens' that pose the greatest threat to human health, calling for the development of more effective therapeutic strategies [2]. Vaccination has proven to be the most cost-effective preventative measure against infectious diseases, and functional vaccines against these 'priority pathogens' are critical for preventing or halting the escalation of antibiotic resistance.

Antigens are the most important components within a vaccine, and immunogenic antigens are crucial for the development of a successful vaccine [3]. Pure proteins or polysaccharides are generally not well recognized by the immune system, and require the addition of carrier systems or adjuvants to improve immunogenicity. Numerous substances with acceptable adverse effects have been tested to improve the immunogenicity of proteins or polysaccharides, but few of them have been clinically approved as adjuvants [4,5]. Among them, alum has been the most successful and widely used adjuvant during the past century [6]. The mechanisms whereby alum improves immunogenicity include, but are not limited to, antigen uptake, antigen depot, and pyrin domain containing 3 protein (NLRP3) inflammasome activation [7]. The identification of substances with adjuvant activity and the elucidation of their mechanism of action are essential for vaccine development.

Pore-forming toxins (PFTs) are produced by numerous pathogenic bacteria to promote their growth and dissemination [8]. Alpha hemolysin (Hla) is a member of the PFT family secreted by *Staphylococcus aureus* (*S. aureus*), and harbors the ability to activate the NLRP3 inflammasome, which subsequently increases IL-1β and IL-18 secretion by activating procaspase-1 [9]. Hla has been identified as a critical virulence factor in a murine model of *S. aureus* infection, and inflammation was observed in the lungs of rats and rabbits treated with this protein [10,11]. Hla targets alveolar epithelial cells via the protein receptor, a disintegrin and metalloproteinase domain-containing protein 10 (ADAM10) [12], and the lipid receptor phosphocholine [13]. Hla has been tested for use in numerous vaccines against *S. aureus*. Inactivated Hla forms, including Hla^H35A, Hla^H35L [14], and Hla AT62 [15], which lack the transmembrane domain, showed no hemolytic capacity but strong antigenicity. Immunization with serotype 5 capsular polysaccharide conjugated to Hla^H35L reduced the dissemination of *S. aureus* into the blood in a murine model [16]. However, to the best of our knowledge, few studies have focused on the impact of Hla toxoid fusion on the immunogenicity of other proteins, and the underlying mechanisms whereby Hla improves immunogenicity remain unclear.

In this study, to avoid the interference in immune protection of antigens from the same species, PA0833 from *Pseudomonas aeruginosa* (*P. aeruginosa*), which we previously

identified as an effective candidate antigen [17], was chosen to construct a fusion protein with Hla$^{H35A}$. The immunogenicity and protective efficacy of the Hla$^{H35A}$-PA0833 fusion protein, termed HPF, were evaluated in a *P. aeruginosa* lethal pneumonia model. The possible mechanism of action was assessed in *in vivo* and *in vitro* experiments. Finally, the universal efficacy of Hla$^{H35A}$ as a carrier protein for antigen design was determined using the model antigen ovalbumin (OVA), and a newly identified vaccine candidate, GlnH, from *Klebsiella pneumoniae* (*K. pneumoniae*).

## Results

### Immunization with HPF resulted in improved protective efficacy against *P. aeruginosa* pneumonia

To test whether Hla$^{H35A}$ fusion improves the immunogenicity of protein antigens, PA0833 and the fusion protein HPF were constructed, purified, and used to immunize BALB/c mice. Immunization was carried out via intramuscular route three times with each of the proteins, which were formulated with or without alum adjuvant. The PA0833-specific immunoglobulin gamma (IgG) titers and their subtypes were determined. Seven days after the first immunization, PA0833-specific IgGs were detectable in the sera of antigen-immunized mice, with no significant difference. After the second immunization, the PA0833-specific IgG titers were significantly increased in all immunized mice. In particular, mice immunized with antigens formulated with alum showed higher IgG titers than those immunized with antigen alone. HPF elicited a higher PA0833-specific IgG titer than PA0833, both with and without alum adjuvant. The PA0833-specific IgG titer induced by HPF was comparable to that induced by PA0833 with alum, which was approximately 15.8-fold higher than that induced by PA0833 alone. However, the third immunization did not further increase the IgG titers, except for HPF with alum (Fig 1A).

Isotype switch to IgG2a and IgG1 is mediated by Th1 and Th2 immune responses in mice, respectively [18]. IgG1 and IgG2a titers were detected one week after the last immunization. Notably, the IgG1 titer in HPF-immunized mice was significantly higher than that in PA0833-immunized mice, and the addition of alum was also able to increase the IgG1 titer. In contrast, the IgG2 titers were improved by the addition of alum, but Hla$^{H35A}$ fusion did not show such an effect. This result demonstrated that alum and Hla$^{H35A}$ fusion synergistically increased the PA0833-specific IgG1 titers, while the IgG2a titers were increased by alum but not the Hla$^{H35A}$ fusion (Fig 1B).

One week after the last immunization, mice were challenged with $1 \times 10^7$ CFUs of the *P. aeruginosa* strain XN-1, and their survival was monitored continuously for 2 weeks. The result showed that 100% of mice were survived in HPF with alum group, while the PA0833 with alum group showed a survival rate of 33.3%. Meanwhile, the PA0833 and HPF groups showed 20% and 40% of survival, respectively (Fig 1C). These results were consistent with the antigen-specific IgG titers elicited by each of the preparations. The lungs from immunized and control mice were harvested 24 h after sub-lethal infection. Bacterial loads (Fig 1D) in the PA0833 with alum and HPF groups showed no difference, but were significantly lower than those in the control group, while the bacterial burden in the HPF with alum group was dramatically decreased when compared to all other groups. In addition, the histological (Fig 1G) and pro-inflammatory cytokine secretion (Fig 1E, 1F, 1H and 1I) examination of the lungs of immunized mice was consistent with the data on survival and bacterial burden. To further confirm the efficacy of Hla$^{H35A}$, the same assays were repeated in a pneumonia model in male C57BL/6 mice. The results showed improved protective efficacy (S1B Fig), immunogenicity (S1C Fig), complement functional activity (S1D Fig) and Th2 responses (S1E, S1F and S1G Fig),

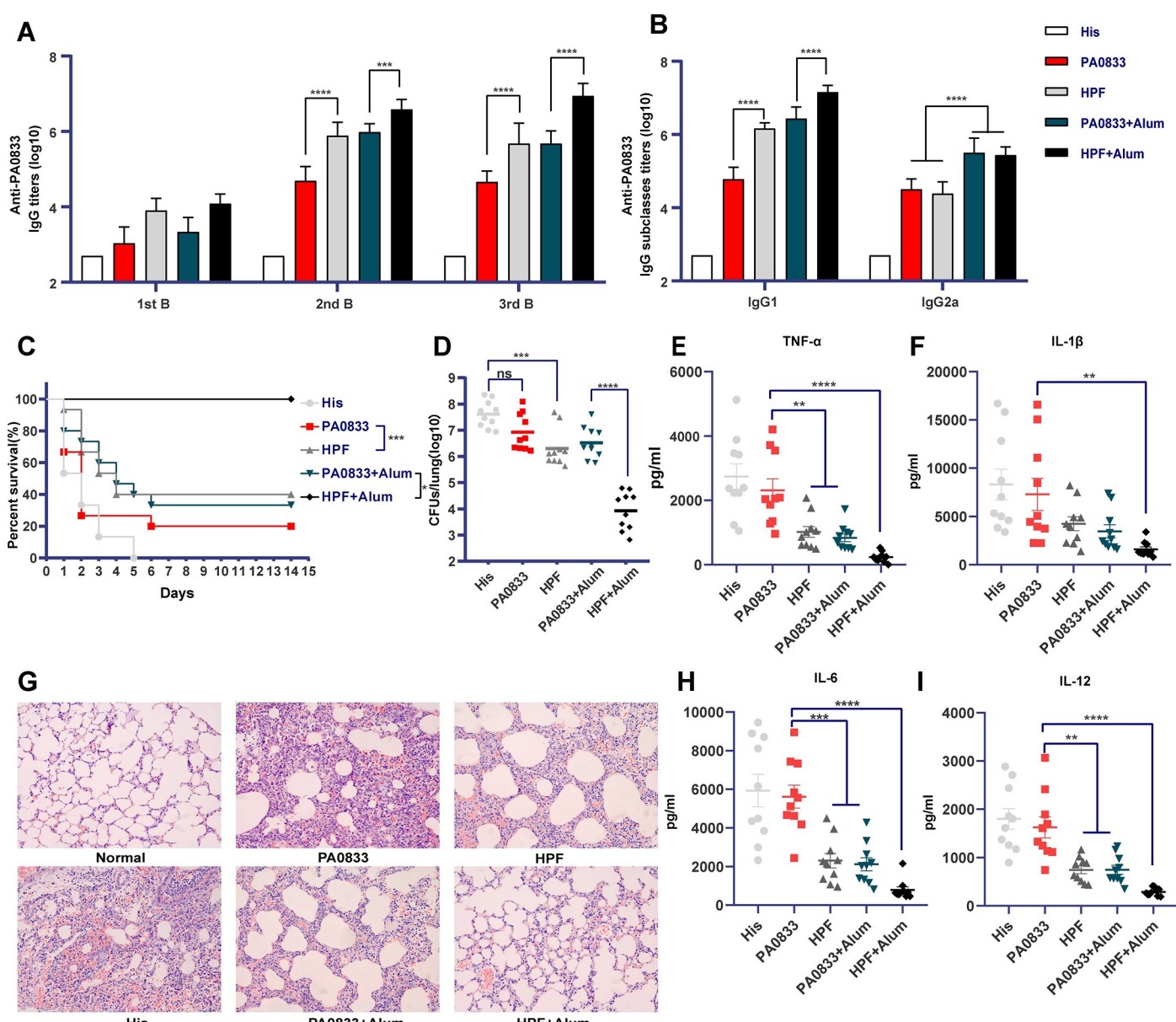

**Fig 1. Hla^H35A fusion resulted in improved immunogenicity and protection efficacy of PA0833 against *P. aeruginosa* lethal pneumonia.** Mice were immunized with His buffer, PA0833 with or without alum, or HPF with or without alum. (**A**) The PA0833-specific IgG titers in sera of immunized mice were determined by ELISA one week after the first, second, and third immunizations. (**B**) The IgG1 and IgG2a titers were assessed one week after the last immunization. Two-way ANOVA, Tukey's multiple comparison test, ***$P$<0.001, ****$P$<0.0001. Data are presented as the mean ± SEM. (**C**) One week after the last immunization, mice were challenged with $1 \times 10^7$ CFUs of *P. aeruginosa* strain XN-1 and monitored continuously for 14 days (n = 15). Gehan-Breslow-Wilcoxon test, *$P$<0.05, ***$P$<0.001. (**D**) Bacterial loads in the lungs were determined 24 h after challenge with a sub-lethal dose of PA XN-1 (n = 10). One-way ANOVA, Tukey's multiple comparison test, ***$P$<0.001, ****$P$<0.0001. (**G**) HE staining of lungs from immunized mice challenged with a sub-lethal dose of PA XN-1. Photomicrographs were taken at 400× magnification. Analysis of the pro-inflammatory cytokines, TNF-α (**E**), IL-1β (**F**), IL-6 (**H**), and IL-12 (**I**) in the lungs of immunized mice 24 h after challenge with $2 \times 10^6$ CFUs of XN-1 (n = 10). One-way ANOVA, Tukey's multiple comparison test, ***$P$<0.001, ****$P$<0.0001. Data are presented as the mean ± SEM.

consistent with that obtained from BALB/c mice. Collectively, these results demonstrated that Hla^H35A fusion and alum formulation enhance the protective efficacy of PA0833, both separately and synergistically.

## Antigen uptake and migration of antigen-positive cells were promoted by the Hla^H35A fusion

After revealing that Hla$^{H35A}$ fusion improves the immunogenicity and protective efficacy of PA0833, the impact of Hla$^{H35A}$ fusion on antigen uptake, which is essential for initiating an immune response, was further investigated. PA0833 and HPF were labeled with Alexa Fluor 488 to enable antigen tracking after immunization. The flow cytometric analysis pipeline is shown in S2 Fig. Antigen-positive cells were determined in muscles and draining lymph nodes (LNs) from all groups (Fig 2A), and antigen-positive cell subsets were gated against the His buffer control group.

In muscles, the number of antigen-positive non-leukocyte (CD45⁻) cells induced with HPF was 4.78 times higher than that with PA0833 alone, and became 13.2 times higher when the antigens were formulated with alum. Absorption of antigens by alum led to a reduction in antigen uptake at 24 h post injection (hpi) but an increase at 72 hpi (Fig 2A and 2B), which suggests that the large size of alum (2–9 μm) [19] halted antigen uptake and degradation by non-leukocyte cells. Formulation with alum resulted in a 2-fold increase in the number of antigen-positive leukocytes (CD45⁺), while Hla$^{H35A}$ fusion resulted in an approximately 4-fold increase at 24 hpi. Thus, the highest number of antigen-positive leukocytes was observed in the HPF with alum group at 72 hpi. Elevation of antigen uptake by leukocytes was observed over time in all immunized groups except for the PA0833 group (Fig 2A and 2B). These results indicated that both Hla$^{H35A}$ fusion and alum promoted antigen uptake at the site of infection.

In draining LNs, the antigen-positive cells homing from the site of injection present the antigens to T or B cells. As shown in Fig 2B and 2C, neither PA0833-positive leukocyte (CD45⁺) cells nor antigen-positive non-leukocyte (CD45⁻) cells were detected at 24 hpi and 72 hpi. In contrast, HPF-positive leukocyte (CD45⁺) cells were detected at 24 hpi, and the frequency of these cells was significantly higher at 72 hpi. This result confirmed that only HPF-positive cells were delivered to draining LNs, which was likely due to Hla$^{H35A}$ fusion.

## The antigen uptake promoted in muscles by Hla^H35A fusion and alum depended on different subsets of infiltrated cells

In view of the elevated level of antigen-specific leukocytes in injected muscles after immunization, the infiltration and antigen uptake efficacy of different leukocyte subsets were further evaluated. CD11c-positive cells were seldom detected in the injected muscles (S2 Fig), which indicated that dendritic cells did not infiltrate into the site of injection or differentiated from infiltrated monocytes. Thus, infiltrated monocytes, macrophages, and neutrophils were further investigated (Fig 3A, 3B and 3C, left panels).

Recruitment of monocytes rapidly vanished in His buffer and peaked in the HPF group; a 3.916-, 1.95-, 1.33-, and 1.88-fold increase in monocytes was detected at 72 hpi, compared to 24 hpi, in the PA0833, HPF, PA0833 with alum, and HPF with alum groups, respectively (Fig 3A). In contrast, antigen-positive monocytes, regardless of the formulation with alum, were dramatically higher in the HPF groups than in the PA0833 groups, both at 24 hpi and 72 hpi, which indicated that the enhancement of antigen uptake with the Hla$^{H35A}$ fusion was mediated by monocytes (Fig 3A). Notably, the percentages of antigen-positive monocytes were similar between HPF with alum and HPF alone at 72 hpi (S3 Fig). The difference in the quantity of antigen-positive cells may be due to a lower recruitment of monocytes induced by alum compared to other types of cells.

The total number of recruited macrophages in each group was similar to that of monocytes. Antigen-positive macrophages reached a peak in the HPF group at 72 hpi, which was remarkably reduced by formulation with alum. Macrophages are mostly derived from infiltrated

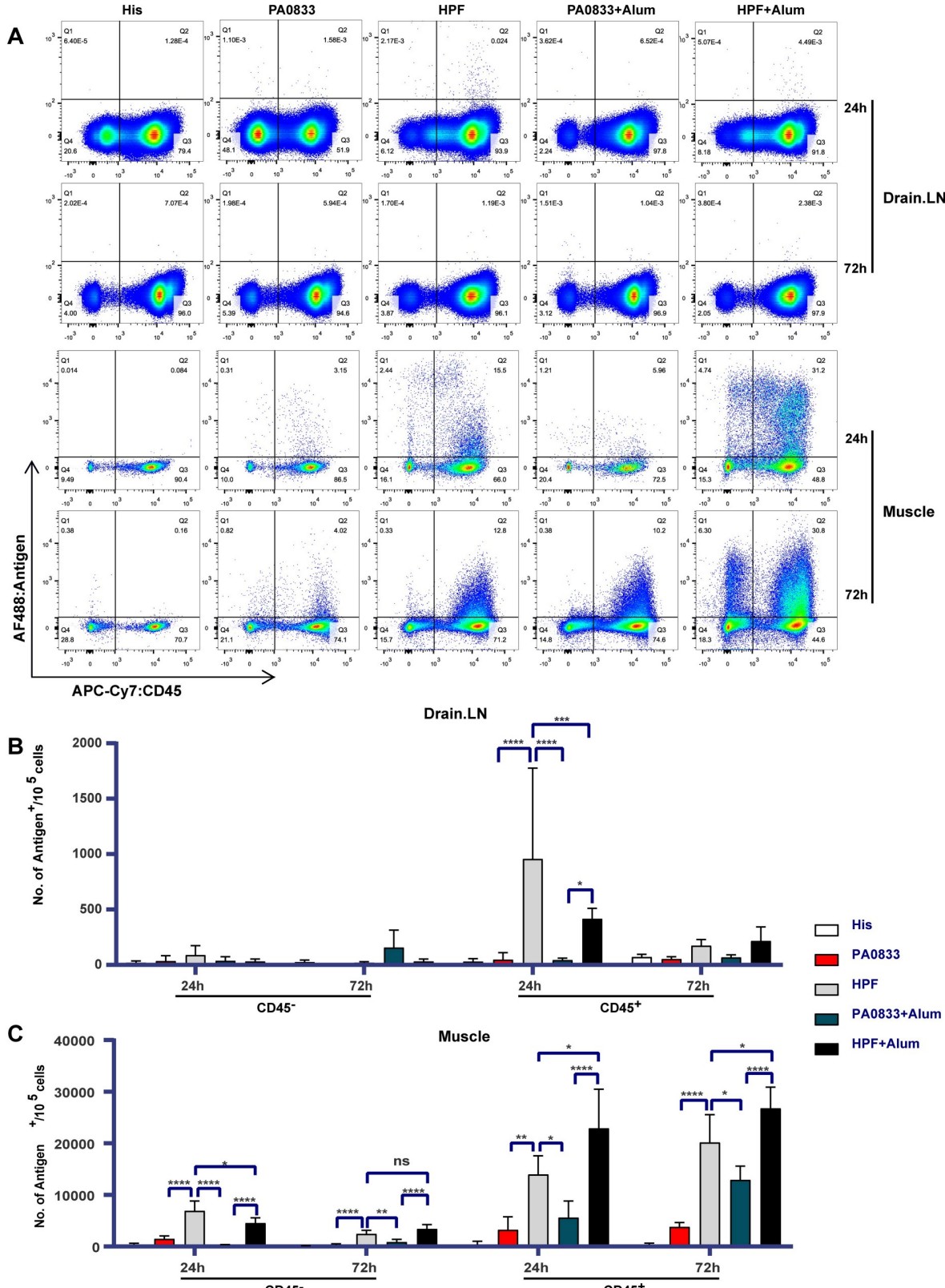

**Fig 2. Antigen-positive cells in muscles and draining lymph nodes (LNs). (A)** Alexa Fluor 488-labeled antigen uptake in the muscle and draining LNs 24 and 72 hours after injection with His buffer, PA0833, HPF, PA0833 + Alum, or HPF + Alum (n = 5). Antigen-positive cells

in the muscles (**B**) and draining LNs (inguinal lymph nodes) (**C**) were detected using a flow cytometer 24 h and 72 h after immunization. Data are shown as the mean ± SEM. Two-way ANOVA, Tukey's multiple comparison test, $^*P{<}0.05$, $^{**}P{<}0.01$, $^{***}P{<}0.001$, $^{****}P{<}0.0001$.

monocytes at the immunization sites [20]. Hence, the significantly fewer antigen-positive macrophages in alum groups at 72 hpi suggested that the formulation with alum did not induce the differentiation of monocytes into macrophages in BALB/c mice (Fig 3B).

Neutrophil recruitment was observed in all immunized mice at 24 hpi, and both Hla[H35A] and alum increased the frequency of neutrophil recruitment. Since neutrophils die within a few hours after the antigen is engulfed [21], neutrophils in the muscles at 72 hpi were newly recruited. The number of neutrophils decreased to normal levels in mice immunized with antigen alone and increased in mice immunized with antigen plus alum, suggesting that alum, but not Hla[H35A], is required for continuous neutrophil recruitment (Fig 3C).

The percentage of antigen-positive monocytes and macrophages increased over time, while that of neutrophils did not differ (S3 Fig). Together, these results suggested that Hla[H35A] fusion

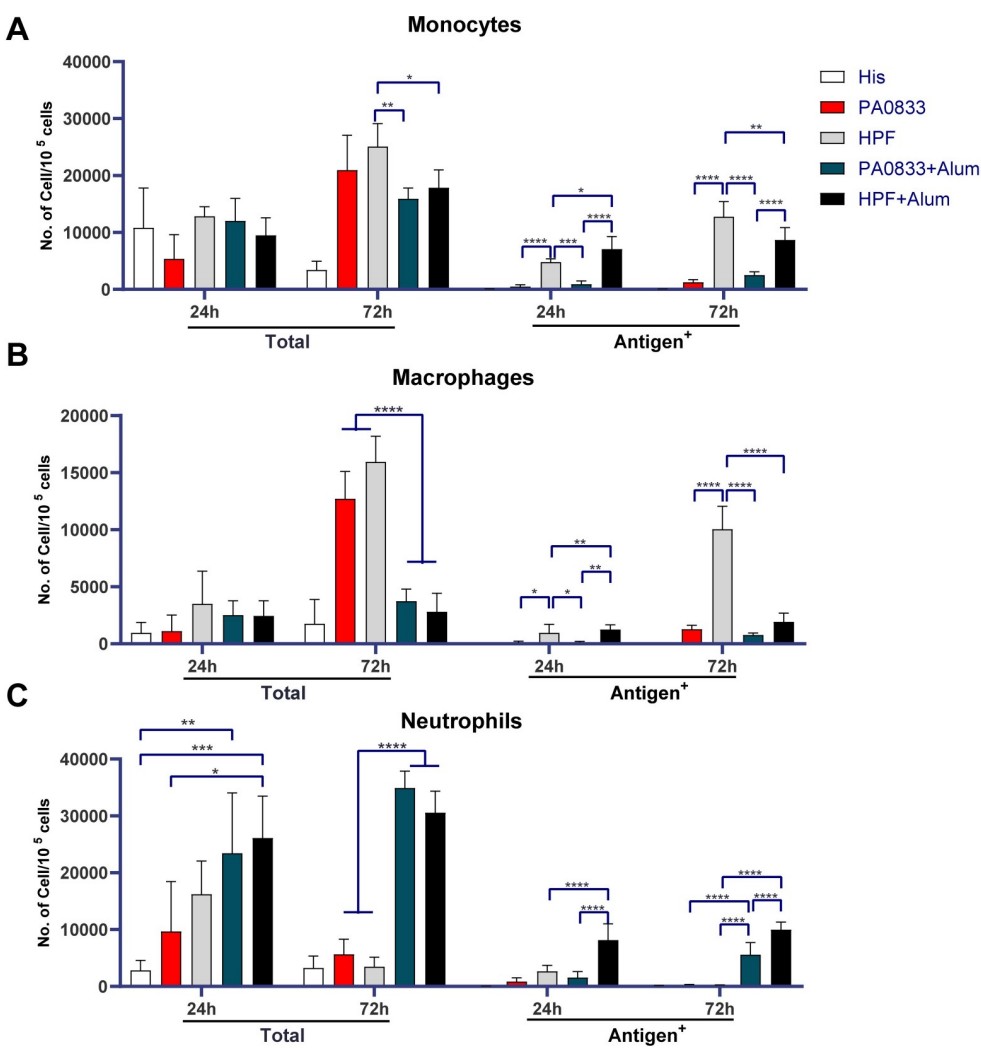

**Fig 3. Monocytes, macrophages, and neutrophils in muscles.** Bars represent the mean and SEM of infiltrated and antigen-positive monocytes (**A**), macrophages (**B**), and neutrophils (**C**) in the muscles 24 h and 72 h after immunization. Two-way ANOVA, Tukey's multiple comparison test, $^*P{<}0.05$, $^{**}P{<}0.01$, $^{***}P{<}0.001$, $^{****}P{<}0.0001$.

mainly improved antigen uptake by monocytes and macrophages, whereas alum improved antigen uptake by neutrophils.

## Hla^H35A fusion improved antigen uptake *in vitro*

Since monocytes and macrophages were mainly responsible for the Hla^H35A-dependent improved antigen uptake, RAW264.7 cells, which are typical antigen-presenting cells, and human alveolar epithelial A549 cells (non-professional antigen-presenting cells) were used for antigen uptake analysis *in vitro*. Considering that antigen uptake is affected by precipitated alum, only antigens without alum were used in *in vitro* experiments. After incubating with the indicated antigens and washing three times with PBS, the cells were lysed and analyzed by immunoblotting. HPF, but not PA0833, was detected in total cell lysates (Fig 4A). A similar phenotype was observed in THP-1 cells, a human monocyte cell line (S4B Fig). However, HPF ceased to increase in the lysates over time, indicating that HPF was attached to the membrane and quickly enzymolysed into polypeptides (Fig 4C). Immunoblotting confirmed that HPF was able to interact with ADAM10, which was identified as the Hla receptor in RAW264.7 and A549 cells, but not in human embryonic kidney-derived 293T cells (Fig 4B). To better detect antigen uptake, PA0833 and HPF were labeled with Alexa Fluor 488. Confocal images showed that the uptake of HPF and PA0833 was time-dependent. In addition, HPF uptake by RAW264.7 cells was significantly higher than that of PA0833. Interestingly, most of the bound HPF was not located at the plasma membrane, but in vesicles (Fig 4D). Similar results were obtained for A549 cells (S4A Fig). The degradation of all the HPF taken by RAW264.7 cells was next examined and found to be a rapid process (Fig 4E). GI254023X, an ADAM10 specific inhibitor, decreased the uptake of HPF by RAW 264.7 cells in a concentration dependent manner (Fig 4F), which suggested that the interaction between ADAM10 and Hla is essential for improved uptake of HPF. These *in vitro* results were comparable to those from the *in vivo* experiments, and suggested that HPF may sequentially be attached to the cell membrane, engulfed into vesicles, and degraded in the cells.

## HPF uptake was macropinocytosis-dependent

Given the efficient uptake of HPF, the host cell machinery implicated in this process was screened using several inhibitors. As shown in Fig 5A, a significant decrease in whole HPF was observed in total cell lysates from RAW264.7 cells treated with amiloride before incubation with HPF, and this inhibition was concentration-dependent. This result was further confirmed by imaging AF488-labeled using confocal microscopy, which showed that the uptake of PA0833 was also inhibited by amiloride (Fig 5B). Amiloride is a specific inhibitor of macropinocytosis, which internalizes a large quantity of plasma membrane and non-selectively uptakes soluble antigens. In contrast, ATP depletion by 2-DG, inhibition of dynamin by dynasore, inhibition of ATPases by vanadate, and inhibition of cholesterol synthesis by lovastatin did not affect whole HPF uptake (S5 Fig). Together, these data demonstrated that Hla^H35A fusion improved antigen uptake mainly through the concurrent internalization of large amounts of plasma cell membrane via macropinocytosis.

## Gene expression in response to HPF and PA0833 administration

In view of the remarkable antigen uptake improvement achieved with the Hla^H35A fusion, we performed transcriptome analysis of RAW264.7 cells treated with His buffer, PA0833, and HPF. The top 50 differentially expressed genes are presented in a heatmap (Fig 6A). Fourteen differentially expressed genes were validated by real-time qPCR using primers listed in S1 Table, and 12 of them are shown. Notably, fusion with Hla^H35A greatly improved the

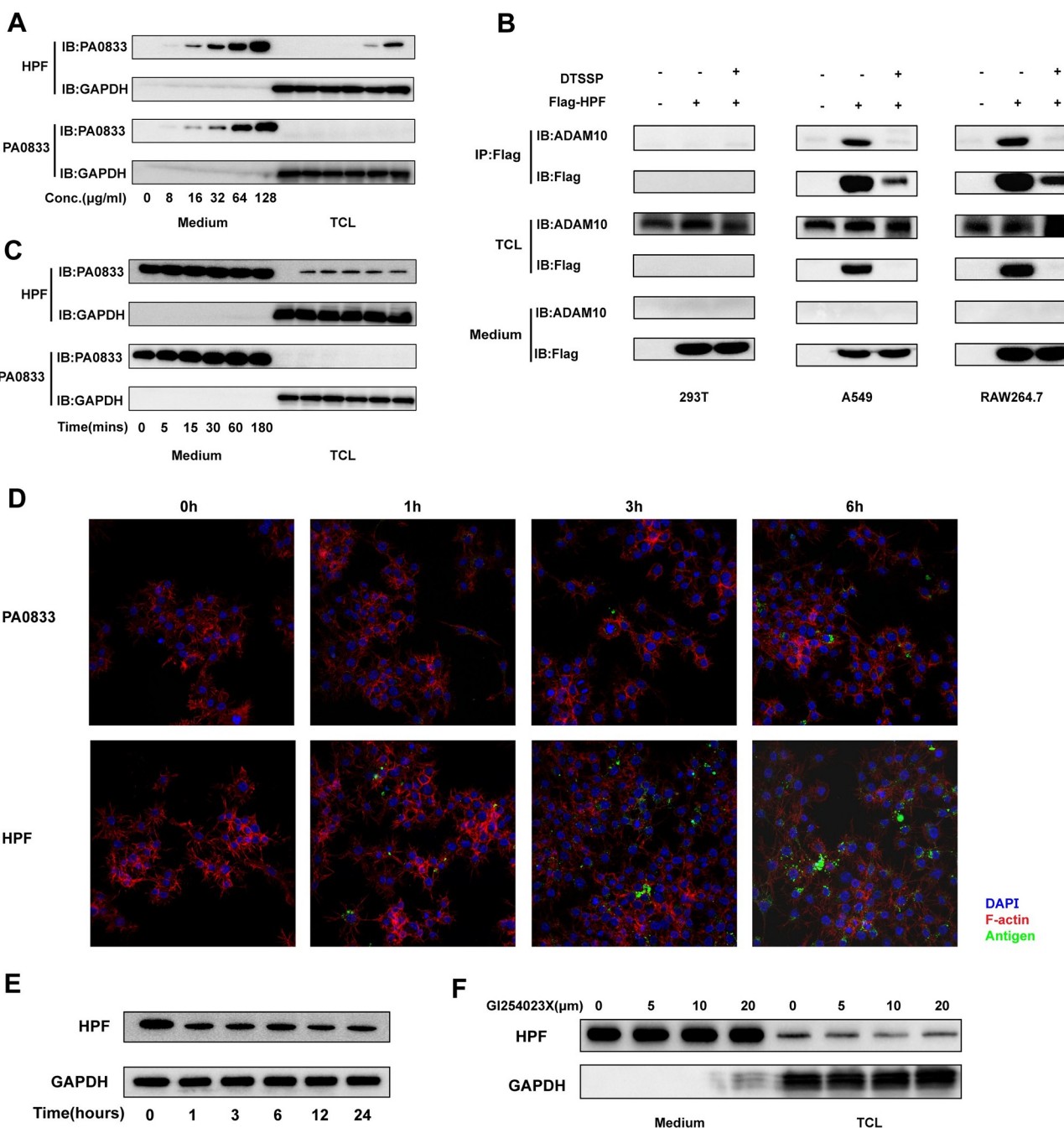

**Fig 4. Antigen uptake by monocytes *in vitro*.** Immunoblotting examining PA0833 and HPF uptake by RAW264.7 cells at different antigen concentrations (**A**) and incubation times (**B**). (**C**) Co-immunoprecipitation and immunoblotting of 293T, A549, and RAW264.7 cells treated with 128 μg/ml of Flag-HPF afterwashing three times with PBS, 0.5 mM 3,3'-dithiobis (sulfosuccinimidyl propionate) was added. (**D**) Confocal imaging of Alexa Fluor 488-labeled PA0833 and HPF uptake by RAW264.7 cells. (**E**) After 3 h of incubation, degradation of HPF was determined by immunoblotting. (**F**) After 16 h pre-treatment of indicated GI254023X, HPF uptake by RAW264.7 cells was measured by immunoblotting.

transcription of pro-inflammatory cytokines, including *il-1α*, *il-1β*, and *il-6*, and type I interferon, *ifn-β* (Fig 6B, 6C, 6D and 6E). The transcription of chemokines, *cxcl2*, *ccl2*, *ccl17*, and *ccl22*, was further enhanced by HPF (Fig 6F, 6G, 6H and 6I). In addition, HPF triggered a higher *cd40* transcription level (Fig 6J), indicating the activity of antigen-presenting cells. *tslp*,

**A**

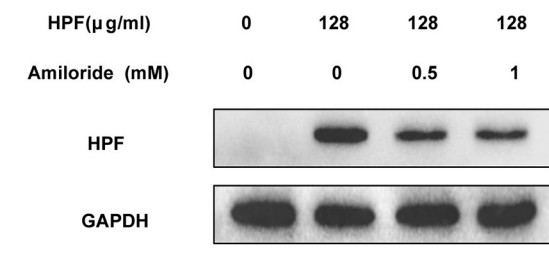

**B**

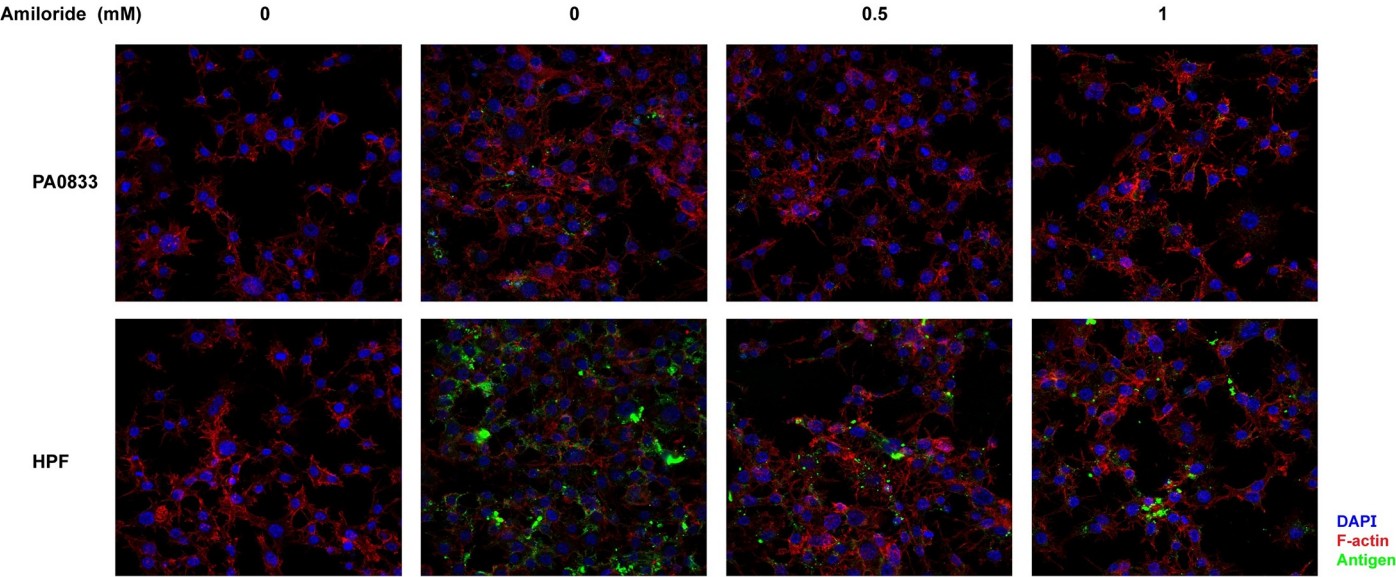

**Fig 5. Antigen uptake of HPF is micropinocytosis-dependent.** HPF (**A**) and Alexa Fluor 488-labeled PA0833 and HPF (**B**) uptake by RAW264.7 cells pretreated with 0, 0.5, and 1 mM amiloride for 1 h. Complete HPF levels were determined by immunoblotting. Alexa Fluor 488-labeled PA0833 and HPF appear green in the confocal images.

a novel IL-7-like cytokine that activates TSLPR+ DCs and plasmacytoid DCs to induce functional Th2, regulatory T (Treg), and human T follicular helper (Tfh) cells [22], was elevated (Fig 6K). The increased transcription of *ptgs2* indicated a higher inflammation level (Fig 6L). Leukemia inhibitory factor (*lif*) has the capacity to induce terminal differentiation of leukemic cells [23]. Thus, the higher *lif* induced by HPF suggested efficient differentiation of RAW264.7 cells (Fig 6M).

To gain a deeper insight into the function of the differentially expressed genes, we performed KEGG pathway annotation and enrichment analysis. The significantly upregulated genes were enriched in 30 pathways (P < 0.05) and are partially shown in S6 Fig. The results clearly show that the TNF (S7A Fig), and IL-17 (S7B Fig) as well as cytokine-cytokine receptor interaction (S7C Fig) pathways were largely affected. In conclusion, Hla$^{H35A}$ fusion induced higher transcription levels of immune-associated genes.

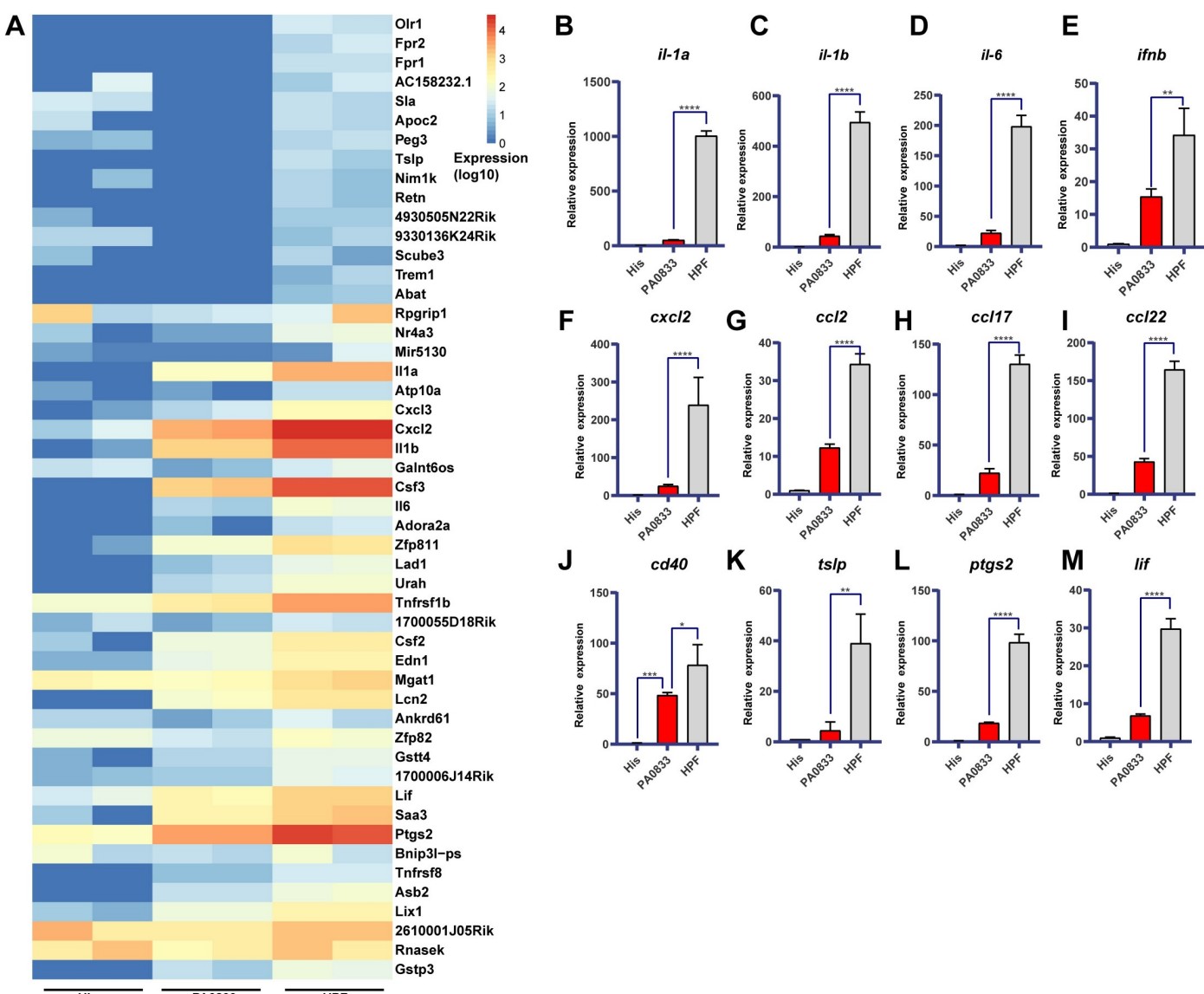

**Fig 6. Transcriptome analysis of RAW264.7 cells treated with PA0833 and HPF. (A)** Heatmap of 50 upregulated genes sorted by high Log Fold Change (HPF versus PA0833). Red color means high gene expression. **(B)** RT-qPCR validation of selected differentially expressed genes. After treatment with His, PA0833, or HPF for 3 h, mRNA transcription in RAW264.7 cells was examined by RT-PCR (n = 4). Bars represent the mean ± SEM. One-way ANOVA, Tukey's multiple comparison test, *$P$<0.05, **$P$<0.01, ***$P$<0.001, ****$P$<0.0001.

## Hla^H35A functions as a universal carrier protein that improves the immunogenicity of protein antigens

After confirming the efficacy of Hla$^{H35A}$ in improving the immunogenicity of PA0833, we wanted to determine whether Hla$^{H35A}$ could be used as a universal carrier protein for antigen design. Thus, the model antigen OVA and GlnH, a candidate antigen that we recently identified from *K. pneumoniae*, were used to construct fusion proteins with Hla$^{H35A}$, and the resulting fusion proteins were named HOF and HGF, respectively. These proteins were used to immunize mice, and the sera were collected to determine the antigen-specific IgG titers and the subtypes, as described previously. After the first boost, similar IgG titers were induced by immunization with either HGF or GlnH with alum, which were approximately 154.8-fold

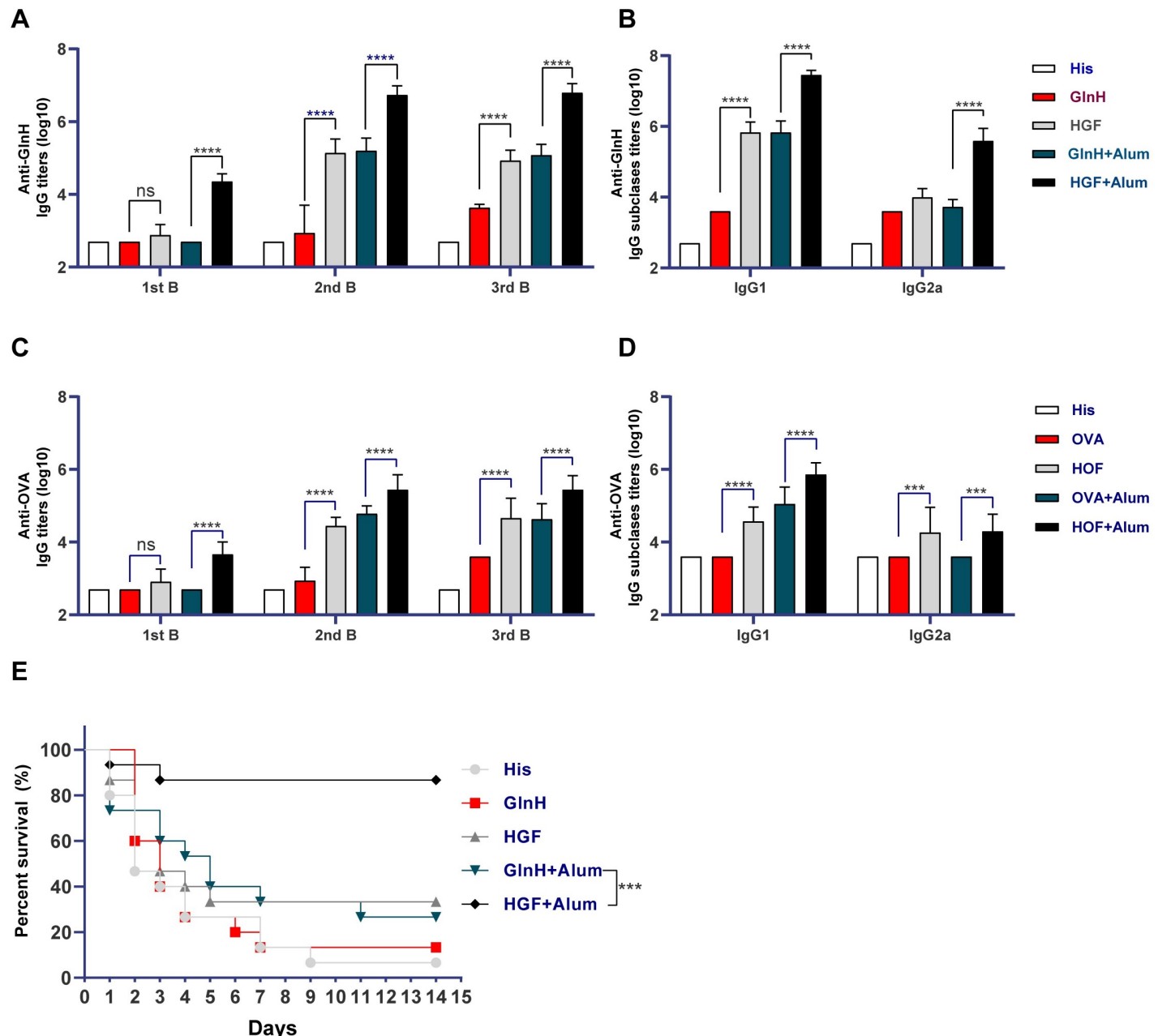

**Fig 7. Hla^H35A fusion improves the immunogenicity of other protein antigens.** Mice were immunized with His buffer, GlnH, OVA, HGF, or HOF with or without alum. **(A)** Titers of GlnH-specific IgGs were determined by indirect ELISA after the first, second, and third immunizations. **(B)** GlnH-specific IgG1 and IgG2a titers were determined after last immunization. **(C)** Titers of OVA-specific IgGs were assessed by indirect ELISA after the first, second, and third immunizations. **(D)** OVA-specific IgG1 and IgG2a titers were evaluated after last immunization. (n = 10). Two-way ANOVA, Tukey's multiple comparison test, ***$P<0.001$, ****$P<0.0001$. **(E)** One week after the last immunization, mice were challenged with $3 \times 10^7$ CFUs of the *K. pneumoniae* strain YBQ and monitored continuously for 14 days (n = 15). Log-rank (Mantel-Cox) test, *$P<0.05$.

higher than that of GlnH alone and 34-fold lower than that of HGF with alum. However, the third immunization did not further improve the IgG titers (Fig 7A). The GlnH-specific IgG1 titers were enhanced by fusion with Hla^H35A, indicating that a significant IgG1 antibody response was elicited by fusion with Hla^H35A (Fig 7B). Survival data from a *K. pneumoniae* lethal pneumonia model were consistent with the observed antibody titers (Fig 7E). Similarly,

fusion with Hla$^{H35A}$ also improved anti-OVA total IgG and IgG1 titers (Fig 7C and 7D). These results are consistent with those observed for PA0833. Altogether, these findings demonstrate that fusion to Hla$^{H35A}$ may be a feasible strategy for protein antigen design with the aim of enhancing the immunogenicity of the selected antigens.

## Discussion

The use of carrier proteins has been proven to be a successful and effective strategy for improving the immunogenicity of polysaccharides, and several polysaccharide-protein conjugate vaccines have been approved for clinical administration [24]. Recently, two pneumococcal vaccines were licensed, in which the immunogenicity of capsular polysaccharides is greatly enhanced by conjugation to a non-toxic variant form of diphtheria toxin [25]. Similarly, promotion of immune protection by Hla$^{H35L}$ conjugated to *S. aureus* serotype 5 capsular polysaccharides indicated that Hla is a promising carrier protein for polysaccharide-conjugated vaccines, although its mechanism of action was not fully elucidated [16]. In the current study, we show that Hla$^{H35A}$ was also capable of improving the immunogenicity and immune protective efficacy of PA0833. Moreover, its efficacy was validated in two other antigens; thus, Hla$^{H35A}$ may serve as a universal carrier protein for antigen design.

Antigen uptake is the first and most critical step in vaccine priming [26]. Here, we present results on antigen uptake by cells recruited into the muscle and antigen delivery to draining lymph nodes. Surprisingly, we did not detect DCs, which are conventionally regarded as the most efficient antigen presenting cells [27], at the site of injection. In contrast, monocytes, neutrophils, and macrophages were recruited to the injected muscles and were responsible for antigen uptake. The continuous recruitment of neutrophils and uptake of antigens by neutrophils was mediated by alum, whereas the uptake of antigens by monocytes and macrophages was attributed to the Hla$^{H35A}$ fusion. Meanwhile, only HPF-positive cells were detected in draining LNs, which suggested that Hla$^{H35A}$ fusion was essential for the migration of antigen-positive cells.

Size and special physicochemical properties are the factors that determine the immunogenicity of a given antigen [28–30]. Prevalent strategies for enhancing the immunogenicity of antigens focus on increasing the size by presenting large particles consisting of self-assembling proteins or peptides [31–34]. Since Hla$^{H35A}$ lose the ability to form a heptamer as its wild type protein [35], the size of HPF is not significantly increased compared to that of PA0833. Hence, the improvement in the immune response caused by Hla$^{H35A}$ fusion was not dependent on an increase in the size of PA0833, but on the special physicochemical properties. The most outstanding feature of pore-forming toxins is their ability to bind to the host cell membrane by protruding their transmembrane domains into the cell host membrane [36]. Inactivated Hla$^{H35A}$ retains the ability to bind to the host cell membrane, but its cytotoxicity is disrupted [35]. In this study, HPF was located at intracellular vesicles, and amiloride efficiently impaired the uptake of HPF by RAW264.7 and A549 cells, indicating that a large amount of the HPF that attached to the cell membrane was internalized by macropinocytosis. ADAM10, which is highly expressed in monocytes and macrophages [37], is a high-affinity host cell receptor for Hla. An H35 single amino acid substitution mutation in Hla and its fusion to a GST tag (a 26 kDa protein) did not impair the binding of Hla to ADAM10 [12], and this was validated by immunoblotting in this study, which demonstrated an interaction between ADAM10 and HPF. Thus, we propose that the enhancement of antigen uptake caused by Hla$^{H35A}$ fusion may be partially due to the targeting of these antigen-presenting cells.

ADAM10, which is responsible for the cleavage of Notch and the regulation of subsequent pathways [38], is indispensable in antigen presenting cells-mediated Th2 immune responses

[39]. In this study, the IgG1 titer was higher when PA0833 was fused with Hla[H35A], suggesting that a stronger Th2 immune response was induced. These results were further confirmed by the higher transcription levels of chemokines associated with a Th2 profile, such as CCL17 and CCL22 [40]. Together, these results demonstrate that the higher immunogenicity elicited by HPF may be ADAM10-dependent. Furthermore, ADAM10 also plays an important role in the immune system, including T cell development [41], humoral immune responses, and marginal zone B cell development [42], which may also contribute to the enhancement of the immune response caused by Hla fusion. Further studies are required to investigate these mechanisms.

The virulence factors of hyper-virulent bacterial strains are not conserved but are immunogenic [43,44]. Most components encoded by the core genome are non-immunogenic and highly conserved, but only a part of them are accessible for antibody recognition [45,46]. The question is how to combine the advantages of non-conservative but highly immunogenic components from hyper-virulent strains, and non-immunogenic but highly conservative components from the core genome. If both of the components are proteins, the fusion protein strategy may provide a feasible solution to this question.

In conclusion, we found that Hla[H35A] fusion efficiently improves the immunogenicity and protective efficacy of PA0833. First, Hla[H35A] fusion enhanced antigen uptake by monocytes and macrophages at the site of injection, and this was dependent on macropinocytosis. Second, the migration of antigen-positive cells to LNs was improved upon Hla[H35A] fusion. Finally, Hla[H35A] induced a stronger Th2 response because of its interaction with the receptor ADAM10. We also demonstrated that this strategy could be easily applied to other protein antigens, suggesting that Hla[H35A] could serve as a universal carrier for antigen design.

## Materials and methods

### Ethics statement

All animal experiments were approved and carried out according to the guidelines of the Animal Ethical and Experimental Committee of the Army Medical University (Chongqing, Permit No. 2011–04). Mice were anesthetized using isoflurane and sacrificed with $CO_2$ to reduce suffering and ensure humane killing.

### Inhibitors

For antigen uptake inhibition, sodium orthovanadate (Selleck), 2-deoxy-D-glucose (Selleck), lovastatin (Selleck), latrunculin B (Adipogen), dynasore (MedChemExpress), GI254023X (Selleck), and amiloride (Selleck) were used at the indicated concentrations.

### Protein expression, purification, and labeling

The model antigen OVA was purchased from Sangon Bioteh Co., Ltd. The DNA sequences encoding PA0833[87-237], Hla[H35A]-PA0833[87-237] (HPF), GlnH, Hla[H35A]-GlnH, and Hla[H35A]-OVA were optimized for expression in *E.coli*, synthesized, and cloned into an expression vector derived from the pGEX-6p-1 plasmid (Novagen) by Sheng Gong Biological Engineering (Shanghai, China) via BamH1 and XhoI. BL21(DE3) cells were transformed with these expression plasmids and induced with 0.2 mM isopropyl-β-d-thiogalactopyranoside (IPTG) at 16°C overnight for the expression of recombinant proteins. Then, the cells were harvested and homogenized using a nano-homogenize machine (ATS Engineering) and the GST-tagged protein was captured with glutathione-Sepharose. The GST tag was cleaved by PreScission Protease (GE Healthcare) after washing with non-specific binding proteins. Protein purity was determined using sodium dodecyl sulfate-polyacrylamide gel electrophoresis (SDS-PAGE,

S1A Fig). Flag-tagged PA0833 and HPF were constructed and purified following the same protocol. PA0833 and HPF were labeled with Alpha Fluor 488 NHS Ester (AAT Bioquest) following the manufacturer' s protocol [27].

## Animals, immunizations, and pneumonia model

Female BALB/c mice and male C57BL/6 mice (6 to 8-week-old) were purchased from HUNAN SJA LABORATORY ANIMAL CO., Ltd. Proteins were diluted with 10 mM histidine buffer (pH 6.0) and formulated with the alum adjuvant (Pierce) at a ratio of 1:1 (v:v). Mice (n = 10) were immunized intramuscularly with 100 μL of a mixture containing 50 μg of each protein on days 0, 14, and 21. Mice immunized with an equal volume of PBS plus adjuvant or 50 μg of each protein without adjuvant were used as controls. The clinical strains of *P. aeruginosa* XN-1 and *K. pneumoniae* YBQ were harvested at $OD_{600nm} \approx 0.6$, and adjusted to the final concentration. After anesthetization with pentobarbital sodium, mice were intratracheally injected with a lethal dose of strain XN-1 ($1.0 \times 10^7$ CFUs, BALB/c mice; $3.0 \times 10^7$ CFUs, C57BL/6 mice) or strain YBQ ($3.0 \times 10^7$ CFUs), and survival in each group was monitored continuously for 14 days. For bacterial burden, histopathology, and cytokine analysis, mice were infected intratracheally with $3.0 \times 10^6$ CFUs of strain XN-1.

## Serum bactericidal assays

Serum bactericidal assays were performed as described previously [47,48]. Briefly, *P. aeruginosa* XN-1 were harvested at $OD_{600nm} \approx 0.6$ and adjusted to the final concentration. 10 μl of *P. aeruginosa* XN-1($2 \times 10^5$ CFUs/ml) was added to 90 μl of freshly thawed, undiluted serum, and incubated at 37˚C with gentle rocking (20 rpm). 50μl of the mixture was plated onto LB agar plates to determine the bacterial clones.

## ELISPOT

ELISPOT assays to detect interferon γ (IFNγ) and IL-4 released by splenocytes were performed as previously described [27]. Splenocytes from C57BL/6 mice immunized or not were stimulated with PA0833 (50μg/ml) for 60 hours. The detection was performed according to the manufacturer's instructions (BD, USA).

## Bacterial burden and histopathology

Mice were sacrificed 24 h after challenge with a sub-lethal dose of *P. aeruginosa* XN-1 and the lungs were fixed with 4% paraformaldehyde, dissected into 4 mm thick sections, and then stained with hematoxylin and eosin for microscopic examination. To determine bacterial burden, the lungs of infected mice were homogenized in 1 mL cold PBS with a Dounce Homogenizer on ice. The lung homogenate was serially diluted in sterile PBS at a ratio of 1:10 (v/v), and 50 μL of diluted homogenate was added to Luria-Bertani (LB) plates. After overnight incubation at 37˚C, the colonies were counted. The results for a mouse are given as the mean of the countable plate.

## Antibody titers

For antibody titer determination, 96-well Stripwel microplates (Corning) were coated with 0.4 μg per well of OVA, GlnH, and PA0833 diluted in carbonate buffer (pH 9.4). The plates were blocked with 1.5% bovine serum albumin (BSA) and incubated with serial dilutions of serum from immunized mice starting from 1:1000, followed by incubation with HRP-labeled goat anti-mouse IgG, IgG1, or IgG2a at dilution 1:5000. Then, a 3,3',5,5'- tetramethylbenzidine

(TMB) substrate solution was added to the wells, and the reaction was terminated with 2 M $H_2SO_4$. The optical density at 450 nm ($OD_{450}$) was measured and the titer of each sample was defined as the highest dilution giving an $OD_{450}$ value higher than 2.1 times that of the negative control.

## Measurement of cytokine production

Twenty-four hours after infection, the lungs of each mouse were homogenized in PBS and the concentrations of pro-inflammatory cytokines, including TNF-α, IL-1β, IL-6, and IL-12, were determined using mouse Quantikine ELISA kits (R&D Systems, USA) according to the manufacturer's instructions.

## Preparation of single cell suspensions from muscles and draining lymph nodes

The preparation of single cell suspensions was performed as previously described [6]. Briefly, the limb muscles and inguinal lymph nodes of immunized mice were harvested into cold Hank's balanced salt solution (HBSS), dissected into fragments, and digested with type II collagenase (Beyotime). The cell suspension was centrifuged, resuspended and mechanically ground through 70 μm cell strainers (Sangon) before staining with fluorescently labeled antibodies for FACS analysis.

## Flow cytometry

After blocking with 2% rat serum, cells were stained for APC-Cy7 rat anti-mouse CD45 (BD), BV421 hamster anti-mouse CD11c (BD), PerCP-Cy5.5 rat anti-mouse I-A/I-E (BD), PE-Cy7 rat anti-mouse Ly-6C (BD), BV510 rat anti-mouse Ly-6G (BD), APC rat anti-mouse F4/80 (BD), PE anti-mouse/human CD11b (Biolegend), and Fixable Viability Stain 700 (BD). The stained cells were analyzed using BD FACS Canto II and DIVA software.

## Cell culture, western blotting, and co-immunoprecipitation

RAW264.7 and A549 cells were cultured in Dulbecco's modified Eagle's medium (DMEM) supplemented with 10% fetal bovine serum (FBS). THP-1 cells were cultured in RPMI 1640 medium supplemented with 10% FBS. After treatment with the indicated concentrations of PA0833 or HPF, cells were washed three times with PBS and lysed. Flag-HPF was immunoprecipitated using anti-Flag M2 magnetic beads (Millipore). For protein detection, immunoblotting was performed with anti-PA0833 rabbit antisera, anti-ADAM10 (Abcam), rabbit anti-Flag (Millipore) as primary antibodies, and anti-rabbit IgG secondary antibody coupled with HRP (ZSGB-Bio) as the secondary antibody.

## Confocal imaging

After treatment with the indicated concentrations of Alpha Fluor 488-labeled PA0833 and HPF, RAW264.7 and A549 cells were fixed with 4% paraformaldehyde and permeabilized with PBS containing 0.3% Triton X-100. Next, the cells were sequentially stained with Alexa Fluor 546 phalloidin and 4',6-diamidino-2-phenylindole (DAPI). Images were collected using a laser scanning confocal microscope (Zeiss).

## RNA extraction, library construction, and sequencing

Total RNA was extracted using TRIzol reagent (Invitrogen, USA) and a miRNeasy Micro Kit (QIAGEN, German) according to the manufacturer's instructions. RNA quality was measured

using an Agilent 4200 TapeStation. RNA quantity was measured using a Qubit 2.0 Fluorometer (Thermo, USA). RNA samples were sent to Genminix Informatic Ltd. (Shanghai, China) for mRNA sequencing on the Illumina HiSeq platform (Illumina, San Diego, California, USA) at 6 Gbps.

## Statistical analysis

Data are shown as the mean ± standard deviation (SD) or the mean ± standard error of the mean (SEM) as indicated. Flow cytometry data were analyzed with Flowjo v10.0 (BD). The structures were analyzed using PyMol2. Survival data were analyzed using Kaplan-Meier survival curves. To determine P-values, log-rank tests, one-way ANOVA with Tukey's multiple comparison test, or two-way ANOVA with Tukey's multiple comparison test were used depending on sample distribution and variation. GraphPad Prism 8.0 (GraphPad Software, Inc.) and Origin 8.0 (Originlab) were used to perform statistical analysis.

## Supporting information

**S1 Fig. Hla$^{H35A}$ fusion improved immunogenicity and protection efficacy of PA0833 against *P. aeruginosa* infection in a C57BL/6 mice pneumonia.** **(A)** SDS-PAGE analysis of PA0833, HPF, GlnH, HGF, OVA and HOF. **(B)** One week after the last immunization, mice were challenged with $3 \times 10^7$ CFUs of *P. aeruginosa* strain XN-1 and monitored continuously for 14 days (n = 10). Log-rank (Mantel-Cox) test, $^*P<0.05$, $^{**}P<0.01$. **(C)** Titers of PA0833-specific IgGs from immunized C57BL/6 mice were determined by indirect ELISA after the first, second, and third immunization (n = 10). Two-way ANOVA, Tukey's multiple comparison test, $^{**}P<0.01$, $^{***}P<0.001$, $^{****}P<0.0001$. **(D)** The *P. aeruginosa* strain XN-1 was tested for survival in serum from different immunization groups. Unpaired student's t test, $^*P<0.05$, $^{***}P<0.001$. ELISPOT analysis of IFN-γ **(E)** and IL-4 **(F)** secretion of splenocytes from different immunization groups. One-way ANOVA, Tukey's multiple comparison test, $^{**}P<0.01$. **(G)** Representative images from ELISPOT analyzes.
(TIF)

**S2 Fig. Analysis pipeline of samples derived from muscles and draining lymph nodes.**
(TIF)

**S3 Fig. Antigen uptake by monocytes, macrophages, and neutrophils in muscles.** Zebra plot of antigen-positive monocytes, macrophages, and neutrophils 24 h and 72 h after injection. Alexa Fluor 488-labeled antigen-positive cells were gated against the His buffer control group.
(TIF)

**S4 Fig. Antigen uptake by A549 cells and human THP-1 cells *in vitro*.** **(A)** Alexa Fluor 488-labeled PA0833 and HPF uptake by A549 cells was captured by confocal microscopy. **(B)** After the indicated incubation, PA0833 and HPF uptake by THP-1 cells was determined by immunoblotting.
(TIF)

**S5 Fig. HPF uptake inhibition screening.** After pretreatment with the indicated concentrations of dynasore **(A)**, 2-deoxy-D-glucose (2-DG)**(B)**, lovastatin **(C)**, and vanadate **(D)**, complete HPF uptake by RAW264.7 cells was determined by immunoblotting.
(TIF)

**S6 Fig. Kyoto Encyclopedia of Genes and Genomes (KEGG) pathway classification of unregulated genes.**
(TIF)

**S7 Fig. Representative significantly enriched KEGG pathways.** Schematic representation of differentially expressed genes enriched in TNF **(A)**, NF-κB **(B)** and cytokine-cytokine receptors interaction **(C)** signaling pathway. Red, white and green color represents up-regulated genes, no difference genes and down-regulated genes, respectively.
(TIF)

**S1 Table. Primers used for transcriptome validation.**
(XLS)

**S1 Data. All numeric values used to generate figures.**
(XLSX)

## Author Contributions

**Conceptualization:** Jin-Tao Zou, Quan-Ming Zou, Jin-Yong Zhang.

**Data curation:** Jin-Tao Zou.

**Formal analysis:** Jin-Tao Zou, Hai-Ming Jing, Yue Yuan.

**Funding acquisition:** Jin-Yong Zhang.

**Investigation:** Jin-Tao Zou, Lang-Huan Lei, Zhi-Fu Chen, Qiang Gou.

**Methodology:** Jin-Tao Zou, Hai-Ming Jing, Yue Yuan, Lang-Huan Lei, Zhi-Fu Chen, Qiang Gou, Qing-Shan Xiong, Xiao-Li Zhang, Zhuo Zhao, Xiao-Kai Zhang.

**Project administration:** Jin-Tao Zou, Hai-Ming Jing, Yue Yuan, Lang-Huan Lei, Hao Zeng, Quan-Ming Zou, Jin-Yong Zhang.

**Supervision:** Hao Zeng, Quan-Ming Zou, Jin-Yong Zhang.

**Validation:** Jin-Tao Zou, Jin-Yong Zhang.

**Writing – original draft:** Jin-Tao Zou, Hai-Ming Jing, Jin-Yong Zhang.

**Writing – review & editing:** Jin-Tao Zou, Quan-Ming Zou, Jin-Yong Zhang.

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
