## [Decision Letter · Decision Letter 0]

16 Apr 2021

Dear Dr. Zhang,

Thank you very much for submitting your manuscript "Pore-forming alpha-hemolysin efficiently improves the immunogenicity and protective efficacy of protein antigens" for consideration at PLOS Pathogens. As with all papers reviewed by the journal, your manuscript was reviewed by members of the editorial board and by several independent reviewers. In light of the reviews (below this email), we would like to invite the resubmission of a significantly-revised version that takes into account the reviewers' comments.

We cannot make any decision about publication until we have seen the revised manuscript and your response to the reviewers' comments. Your revised manuscript is also likely to be sent to reviewers for further evaluation.

Sincerely,

Rachel M McLoughlin, PhD

Associate Editor

PLOS Pathogens

Alan Hauser

Section Editor

PLOS Pathogens

Kasturi Haldar

Editor-in-Chief

PLOS Pathogens

orcid.org/0000-0001-5065-158X

Michael Malim

Editor-in-Chief

PLOS Pathogens

orcid.org/0000-0002-7699-2064

Reviewer's Responses to Questions

**Part I - Summary**

Reviewer #1: In this manuscript “Pore-forming alpha-hemolysin efficiently improved the immunogenicity and protective efficacy of protein antigens”, Zou et al describe the use of a mutated form of the S. aureus protein Hla as a protein carrier for fusion to antigens used for immunization. The data suggests that the fusion proteins, when combined with alum, can offer substantial protection (100% survival and over 3 log reduction in bacterial burden) from infection with P. aeruginosa. The authors go on to show that the Hla-fusion proteins can induce robust antigen-specific IgG responses, in addition to cytokine responses. They show that antigen uptake occurs by monocytes, macrophages and neutrophils all uptaking the antigen. Overall, this is a concise and well-written manuscript with robust data.

Reviewer #2: This study by Zou and colleagues tests a novel hemolysin and pseudomonas antigen fusion protein as a novel vaccine platform. The strength of the study is mainly in the innovation of this construct. The data in mice show increased antibody responses compared to immunization with the pseudomonas antigen alone and also impressive enhancement of protective immunity, but only in the presence of alum. The protective immunity is not improved without alum as shown in Figures 1C and 1D. In addition, to achieve this effect three immunization are necessary. The immunity is not tested against other pseudomonas antigen, only one mouse strain and one sex are tested. This limits the enthusiasm for the potential applicability of this platform. The cellular response studies in muscle and lymph nodes seem mostly dependent on the alum effect. Also, the conclusion that there is an increased Th2 response is only based on slightly increased IgG1 titers.

The relevance of the in vitro studies is limited by the use of tumor cell lines.

Reviewer #3: The authors described a fused protein as potential vaccine candidate. Several fused proteins including chimeric and trimeric vaccines were previously described. The authors did not show the advantage of their fused protein compared to other vaccine candidates against pseudomonas.

**Part II – Major Issues: Key Experiments Required for Acceptance**

Reviewer #1: The authors did not compare their HPF to one of the well-established protein carriers like diphtheria toxin or tetanus toxoid to test if Hla is indeed a superior protein carrier than molecules with known safety profiles. Without these data, it is hard to gauge the potential impact of the presented findings.

The lack of DC recruitment to the injection site is puzzling and may need to be further investigated.

Reviewer #2: Confirmation of the enhancement of protective and Th2 immunity in a different haplotype.

Clearer delineation of the alum dependent responses.

Reviewer #3: -Complement mediated killing has a major role in clearance of Gram negative bacteria including pseudomonas species. The authors did not show how complement functional activity affected before and after immunisation.

-The authors cloned 2 coding sequences in one expression vector.Would you please describe this in details. Did you express the fused protein under control of one promotor or your vector has two different promotors.

-The discussion section is poorly written and the authors did not high light other vaccine candidates against pseudomonas.

**Part III – Minor Issues: Editorial and Data Presentation Modifications**

Reviewer #1: Could the authors use and ADAM10 inhibitor and show that the response to the fusion protein is diminished, therefore directly linking the increased response of using Hla to the binding of ADAM10?

Reviewer #2: (No Response)

Reviewer #3: -The resolution of the figures is not good

- English language editing is also required in some section.

PLOS authors have the option to publish the peer review history of their article (what does this mean?). If published, this will include your full peer review and any attached files.

Reviewer #1: No

Reviewer #2: **Yes: **Stefan Worgall

Reviewer #3: No
---

## [Decision Letter · Decision Letter 1]

24 Jun 2021

Dear Dr. Zhang,

We are pleased to inform you that your manuscript 'Pore-forming alpha-hemolysin efficiently improves the immunogenicity and protective efficacy of protein antigens' has been provisionally accepted for publication in PLOS Pathogens subject to you addressing the outstanding reviewers comments included below.

Best regards,

Rachel M McLoughlin, PhD

Associate Editor

PLOS Pathogens

Alan Hauser

Section Editor

PLOS Pathogens

Kasturi Haldar

Editor-in-Chief

PLOS Pathogens

orcid.org/0000-0001-5065-158X

Michael Malim

Editor-in-Chief

PLOS Pathogens

orcid.org/0000-0002-7699-2064

Reviewer Comments (if any, and for reference):

The figure legends and description of the results should be clear on which mouse strain was used for the respective experiments.

The lack of testing of other promising Pseudomonas proteins should be discussed as a potential weakness of this study

There is still a question of how much improved this platform really is compared to others.

---

## [Editor Report · Acceptance letter]

5 Jul 2021

Dear Dr. Zhang,

We are delighted to inform you that your manuscript, "Pore-forming alpha-hemolysin efficiently improves the immunogenicity and protective efficacy of protein antigens," has been formally accepted for publication in PLOS Pathogens.

Best regards,

Kasturi Haldar

Editor-in-Chief

PLOS Pathogens

orcid.org/0000-0001-5065-158X

Michael Malim

Editor-in-Chief

PLOS Pathogens

orcid.org/0000-0002-7699-2064